



# Water vapor in cold and clean atmosphere: a 3-year data set in the boundary layer of Dome C, East Antarctic Plateau

Christophe Genthon[1], Dana Veron[2], Etienne Vignon[1], Jean-Baptiste Madeleine[1], Luc Piard[3]

[1]Laboratoire de Météorologie Dynamique, Paris, France
[2]Department of Geography and Spatial Sciences, University of Delaware, Newark, USA
[3]Institut des Géosciences de l'Environnement, Grenoble, France

*Correspondence to*: Christophe Genthon (christophe.genthon@cnrs.fr)

**Abstract.** The air at the surface of the high Antarctic Plateau is very cold, dry and clean. In such conditions the atmospheric
moisture can significantly deviate from thermodynamic equilibrium conditions, and supersaturation with respect to ice can
occur. Most conventional humidity sensors for meteorological applications cannot report supersaturation in this environment.
A simple approach for measuring supersaturation using conventional instruments, one being operated in a heated airflow, is
presented. Since 2018, this instrumental setup was deployed at 3 levels in the lower ~40 m above the surface at Dome C on
the high Antarctic Plateau. The 3-year 2018-2020 record  (Genthon et al. 2021) is presented and analyzed for features such as
the frequency of supersaturation with respect to ice, diurnal and seasonal variability, and vertical distribution. As supercooled
liquid water droplets are frequently observed in clouds at the temperatures met on the high Antarctic Plateau, the distribution
of relative humidity with respect to liquid water at Dome C is also discussed. It is suggested that, while not strictly
mimicking the conditions of the high troposphere, the surface atmosphere on the Antarctic Plateau is a convenient natural
laboratory to test parametrizations of cold microphysics predominantly developed to handle the genesis of high tropospheric
clouds. Data are distributed on the PANGAEA data repository at https://doi.pangaea.de/10.1594/PANGAEA.939425
(Genthon et al., 2021).

## 1 Introduction

The surface atmosphere of the high Antarctic Plateau is very cold, with an annual average 3-m air temperature of -52°C,
ranging from -64°C in June to -31°C in January (Genthon et al. 2021). It is cold even in the summer because the permanent
snow cover has a high albedo and reflects much of the incoming solar radiation. The atmosphere is particularly cold in
winter because the sun is low on or below the horizon, while the snow surface efficiently radiates thermal energy through
the dry atmosphere above. In fact, during the polar night, the only external source of heat for the surface and the overlying
atmospheric column is that transported by the atmospheric circulation. Concerning moisture, it is advected inland from the
surrounding oceans including that part of the atmospheric moisture that feeds the ice sheet as snowfall at the surface. It is no
wonder that, considering the distance from the coast and the large temperature gradient from the sources of moisture around
the ice sheet to the deep interior, much of the atmospheric moisture deposits on the way and little is left in the atmosphere



when reaching the high plateau: the atmosphere above the plateau is not only very cold but also very dry. In addition, because the aerosol sources are remote, the atmosphere is also very clean, containing few impurities that could act as condensation nuclei (Herens et al. 2018) and even more as freezing nuclei (Belosi et al. 2014).


It may be expected that the moisture conditions in such a cold and clean atmosphere are unusual, when compared to a more conventional, warmer and less clean atmosphere. Meteorological and climate models struggle to simulate cirrus clouds in a similar environment at high altitudes in the troposphere when using basic parameterization that use a relative humidity threshold where clouds form when the atmosphere becomes supersaturated with respect to liquid water (wrl) for warm

clouds, with respect to ice (wri) in cold clouds (Kärcher and Lohmann, 2002). This has led to improved formulations that take into account the fact that supersaturation is possible in a cold and clean atmosphere (Gettelman et al. (2006), Tomkins et al. (2007)). Although this improvement has sometimes been ignored leading to erroneous recommendations for the required qualities of IPCC models (Genthon et al. 2018), one may expect a similar need for advanced parameterization to reproduce observations of atmospheric moisture at the surface of the high Antarctic Plateau. However, the traditional sensors used to

measure atmospheric moisture are inherently unable to report supersaturation because the instrument itself, and any solid surfaces nearby, act as condensation sites such that when supersaturation occurs in the free atmosphere, the "excess" moisture (moisture exceeding saturation) is deposited on nearby surfaces before reaching the moisture sensing device itself. In Antarctica, this phenomenon is reported by King and Anderson (1999) and further described in Genthon et al. [2017], who also present a simple solution to this measurement challenge employing an arrangement of commercially available

thermohygrometers able to measure and report supersaturation (their figure 1). Measurements taken with this experimental setup over one year at Dome C on the high Antarctic Plateau revealed that, in the surface atmosphere, supersaturation can reach well above 100% wri. This is frequent and in fact the norm rather than an exception (Genthon et al. 2017).









In Genthon et al. (2017), moisture observations were made at only one level, ~3 m above the surface, with this new observational setup that observed  supersaturation. The setup design was later improved (figure 1) for better air circulation and enhanced protection against possible radiation biases, and then 3 new thermohygrometer instrument packages were

deployed at ~3, ~18 and ~42 m above the surface along the ~42-m tower at Dome C [Genthon et al. 2021]. From then on the observing system was unchanged and operating almost continuously, albeit with some operations challenges, providing a quasi-continuous 3-year time-series of the atmospheric humidity profile in the near-surface layers of the high Antarctic Plateau. The observation of the vertical humidity gradient is particularly important because this is the origin of turbulence being able to transport moisture vertically and exchange with surface. Meteorological and climate models cannot explicitly

calculate turbulent fluctuations and correlations and resulting turbulent flux. Instead these models employ bulk formulations to calculate the surface stress and surface turbulent heat and moisture fluxes, with correction functions to account for the fact that the Monin-Obukov theory does not generally apply well in the strongly stable surface layers found on the high Antarctic Plateau (Vignon et al. 2017).



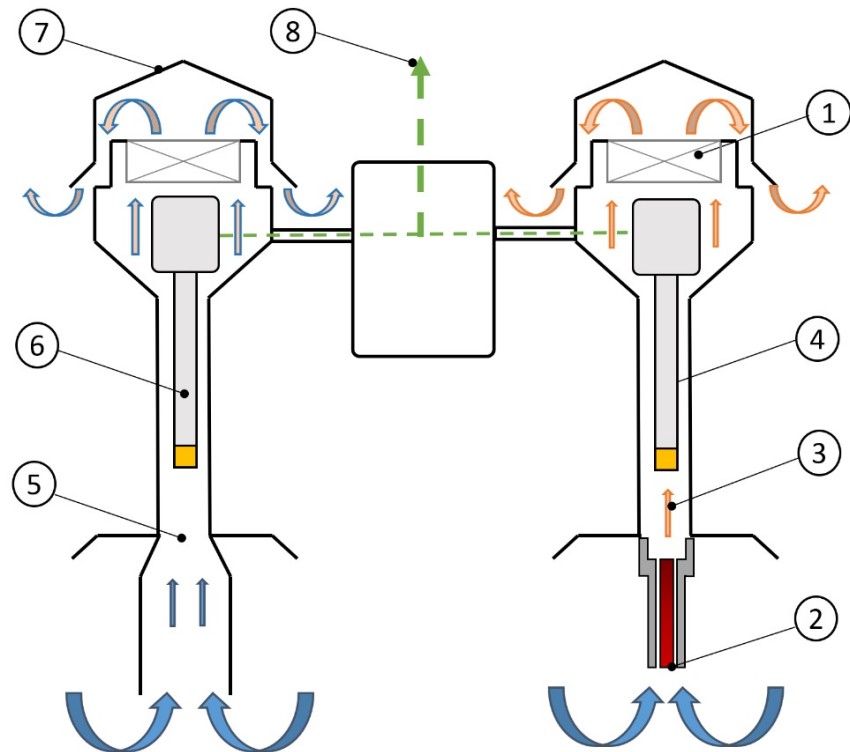

*Figure 1: Schematic of the improved design of Genthon et al. (2017) for measuring atmospheric moisture, even when it is above saturation. Air is aspirated in a ventilated radiation shield (1) through a heated inlet (2). Its relative humidity and temperature (3) are measured with a HMP155 thermohygrometer (4). At the same time, the air is also aspirated through an unheated inlet (5) and ambient temperature measured by a second HMP155 (6) set in a separate shield (7). Data from both HMP155s are collected by a common datalogger (8).*

To test the limits of MO theory and its application for moisture fluxes in meteorological and climate models, direct measurements of turbulent moisture flux are needed. However, the instrumental technology for this in situ measurement is complex and not quite ready for long duration campaigns in the extreme conditions of the high Antarctic plateau. Thus, the data reported here can be used to evaluate the ability of meteorological and climate models to reproduce the vertical moisture gradients used to represent the vertical distribution and mixing of moisture in the boundary layer, but not the fluxes themselves. The data reported here can also be used to verify the performance of parameterizations of cold microphysics implemented in models, for the conditions of the surface at Dome C.



In this paper, we present data and provide limited analyses of 3 years of atmospheric moisture measurements at 3 levels along the lower ~40 m of the atmosphere above the surface, including characterization of supersaturation when it occurs. Analyzes of aspects related to condensation at very cold temperature resulting from supersaturation are left for forthcoming papers. At this time, we consider the humidity data sufficiently new and useful for evaluating current models and paradigms to pass them on to the scientific community. The data are hosted by the PANGAEA archiving and distribution facility where it can be accessed [Genthon et al. 2022]. After the present introduction (Section 1), Section 2 describes the general measurement setting and technical methods for the observations. Section 3 presents the main features of the new humidity data, such as variability, extremes and vertical gradients of moisture content and relative humidity with respect to ice. As liquid water clouds are known to occur well below 0°C in the high troposphere, and have been observed at temperatures below -20°C above Dome C (Ricaud et al. 2020a), section 4 provides a quick look and discussion of saturation with respect to liquid water as calculated from our observations for the surface atmosphere of the Antarctic Plateau. Section 5 provides general conclusions.

## 2. Observation site, instruments and methods

Genthon et al. (2021) presented 10 years of observations of wind and temperature at 6 levels along a 42-m tower at Dome C,123° 21' E, 75° 06' S, 3233 m above sea level. The dome C environment is cold, dry and spatially very homogeneous. Genthon et al. [2021] report temperatures ranging between -80°C to -15°C in the near-surface atmosphere. Figure 2 shows the variations of the equilibrium water vapor pressure over ice in this temperature range, according Goff and Gratch (1946)'s (GG henceforth) empirical analytic adjustment of Clausius-Clapeyron (CC) relations. The GG formulations are not the most recent or accurate such approximation, however it was widely used and reported in the general literature. Differences with other equations are not large enough to affect discussions and conclusions in the present work. The observational time-series are made available in native sensor units, which is relative humidity with respect to liquid for the Vaisala HMP155 thermohygrometer used here, even for temperatures below 0°C. The users of this new dataset can recalculate relative humidity with respect to ice (RHi) and other characteristics of atmospheric moisture using their preferred method.

The water vapor partial pressure (PPW) extends over more than 3 orders of magnitude (figure 2), because the relation between PPW and temperature is near exponential and the range of observed temperature is large. The steep and frequent temperature inversions observed at Dome C (Genthon et al. 2021) cause very low near-surface temperatures while the air can be up to 30°C warmer just 40 m higher in the atmospheric column, adding to the large temperature range due to diurnal and seasonal variability. Correspondingly large saturation pressure vapor ranges result in these settings. Therefore, instruments observing atmospheric moisture in such environment must not only perform at very cold temperatures but also, and consequently, at very low moisture contents and across a large range of humidities. A reference instruments such as the frost point hygrometer as used by King and Anderson (1999) and Genthon et al. (2017) can generally not measure in the coldest



conditions at Dome C. Genthon et al. (2017) reported using a frost-point hygrometer that, although designed specifically for cold environments, could not perform below -55°C, due to limitations of the the Peltier mirror cooling device that could not

operate correctly below this temperature. This issue was not reported by King and Anderson (1999) but temperatures at their site (Halley station) were much less extreme than at Dome C. Versions of Vaisala's Humicap© thin-film capacitive sensors used in the HMP155 thermohygrometer are factory calibrated and validated down to -60°C. Although operating outside factory-calibration range, these sensors provide data well below this lower limit, down to -80°C. According to the manufacturer, the issue at the coldest temperatures is related to a long time response rather than accuracy ((personal

communication). As the moisture content is unlikely to change rapidly at a fixed location and height in the conditions at Dome C, this is not considered a major issue here, yet results reported for the coldest temperatures should be considered with some caution. With the Humicap, relative humidity with respect to liquid water (even below 0°C) is an empirically calibrated function of measured thin-film capacity. According to manufacturer, in the range -40°/-60°C, the accuracy is a function of relative humidit y: ±(1.4 + 0.032 x reading) in % relative humidity with respect to liquid.


The temperatures presented by Genthon et al. (2021) are obtained from several HMP155 instruments, which also report relative humidity with respect to liquid water as mentioned above, set in ventilated radiation shields deployed at 6 levels along the tower. Sensors on the tower were sampled at 30-second intervals. Averages, minima, maxima and variances were calculated over 30-minute periods and stored using a Campbell CR3000 datalogger. Genthon et al. (2021) focus on

temperatures from the HMP155 (and wind from other sensors) but do not yet report on the moisture data because the sensors are limited to measuring up to 100% relative humidity wri, while it is known for the atmosphere to reach well above saturation at Dome C (Genthon et al., 2017).

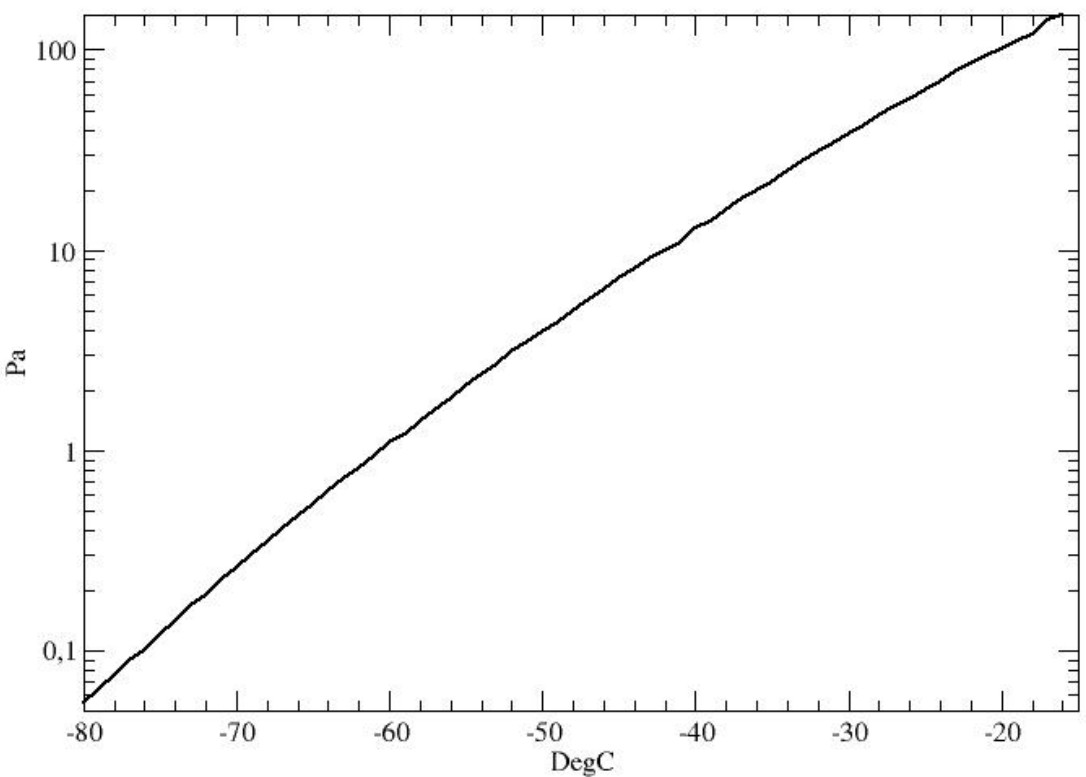

*Figure 2: Calculated saturation pressure vapor (in Pa) with respect to ice as a function of temperature in the range*
*occurring in the surface atmosphere at Dome C, using the Goff and Gratch (1946)approximation of Clausius-Clapeyron*
*equilibrium. Note the logarithmic y-axis.*

## 3. The atmospheric moisture data set

### 3.1. Water – temperature correlation and evidence of supersaturation


Figure 3 shows the correlations between PPW and temperature as reported in the Dome C surface atmosphere by the
conventional instruments (HMP155, 3a) and by the instruments adapted to measure supersaturation if and when it occurs
(3b). The traditional view that the water vapor pressure in the atmosphere cannot reach above saturation would lead to a
distribution of water vapor with respect to temperature capped by CC curve (figure 2). As shown by figure 3a, the standard

HMP155 would support such a view and is thus misleading. The main difference between 3a and 3b is that the standard instrument hardly reports any case of water content significantly above that of CC thermodynamic equilibrium while this is frequently largely above with the adapted instruments. The frequency of supersaturation cases (cases above the CC line) varies as a function of temperature. In the region "below" CC equilibrium, the 2 instruments report similar humidities and dependence with temperature.


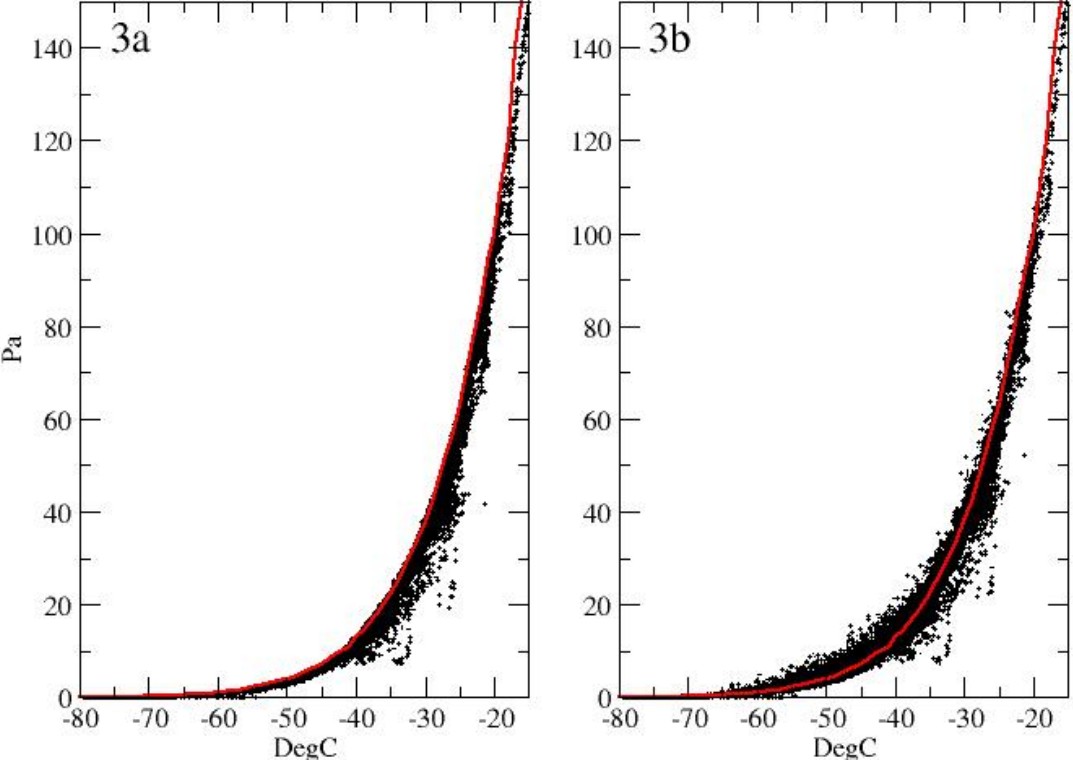

*Figure 3: Observed distribution of water vapor versus temperature for 2018-2020, as reported by standard instruments such as HMP155 (3a) and systems adapted to report supersaturation (3b). The red line shows the Clausius-Clapeyron relation as* 175 *calculated using the Goff and Gratch (1946)formulation, similar to Figure 2 but shown here with a linear y-axis.*

Observations made with standard instruments are misleading each time the atmospheric water content is significantly above saturation. Figure 4 displays the percentage of cases in the time series with supersaturation wri above 105% within given temperature ranges by 5°C interval; supersaturation can occur over the full range of temperatures at Dome C and thus in all 180 seasons and at all levels, with frequency varying with temperature, but it is more frequent in the intermediate temperature



range. The coldest temperatures in the surface atmosphere at Dome C are associated with calm conditions. This may be due to stagnant air slowly cooling radiatively, which should result in supersaturation, or air subsiding from above in association with the general convergence and subsidence on the Antarctic Plateau (Bas et al. (2019), Vignon et al. (2018)). In the latter case, adiabatic warming through subsidence and compression reduces relative humidity at a fixed moisture content, thus

contributing to less frequent supersaturation cases at very cold temperatures. At the warmer end of the distribution are the summer days associated with advection of comparatively warm and aerosol laden air masses from the lower latitudes. For air masses in the middle of the temperature range, adiabatic and radiative cooling combine to increase relative humidity of relatively clean air masses contributing to the frequent occurrence of supersaturation wri.


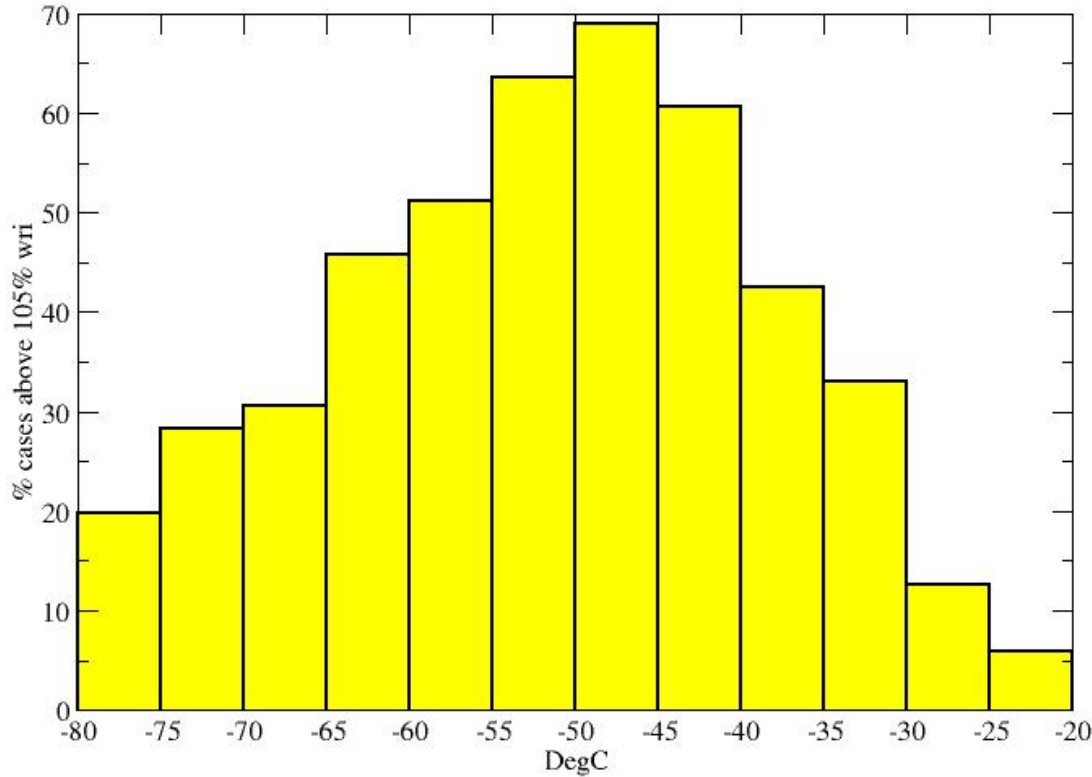

*Figure 4: Percentage of cases with supersaturation wri above 105% from 2018-2020 binned in 5°C from -80 to -20°C.*





Figure 5 presents the frequency distribution of occurrences of relative humidity wri in 10% bins, from 75-85% (centered on

80% on x axis) to 135-145% (140%) for the 3 levels on the tower instrumented with the paired hygrometers. The 18-m level

does not explicitly show at the 2 extremes of the distribution because the corresponding frequencies are the same for the 42-

m level. In the 125-135% range the observations at the 3-m and 42-m levels have the same value. Below the 100% threshold,

the relative humidity is largest near the surface, and then decreases with height. This is consistent with temperature inversion

that  would result in higher relative humidity in the colder layers near the surface, even if the absolute moisture content was

the  same  for  all  levels.  This  does  not  hold  for  higher  values  of  RHi , particularly  at  the  largest  values  of  RHi  in  the

intermediate levels rather than at one of the extremes: the profile of RHi with elevation is not monotonic, a fact which is

further discussed in Section 3.3.

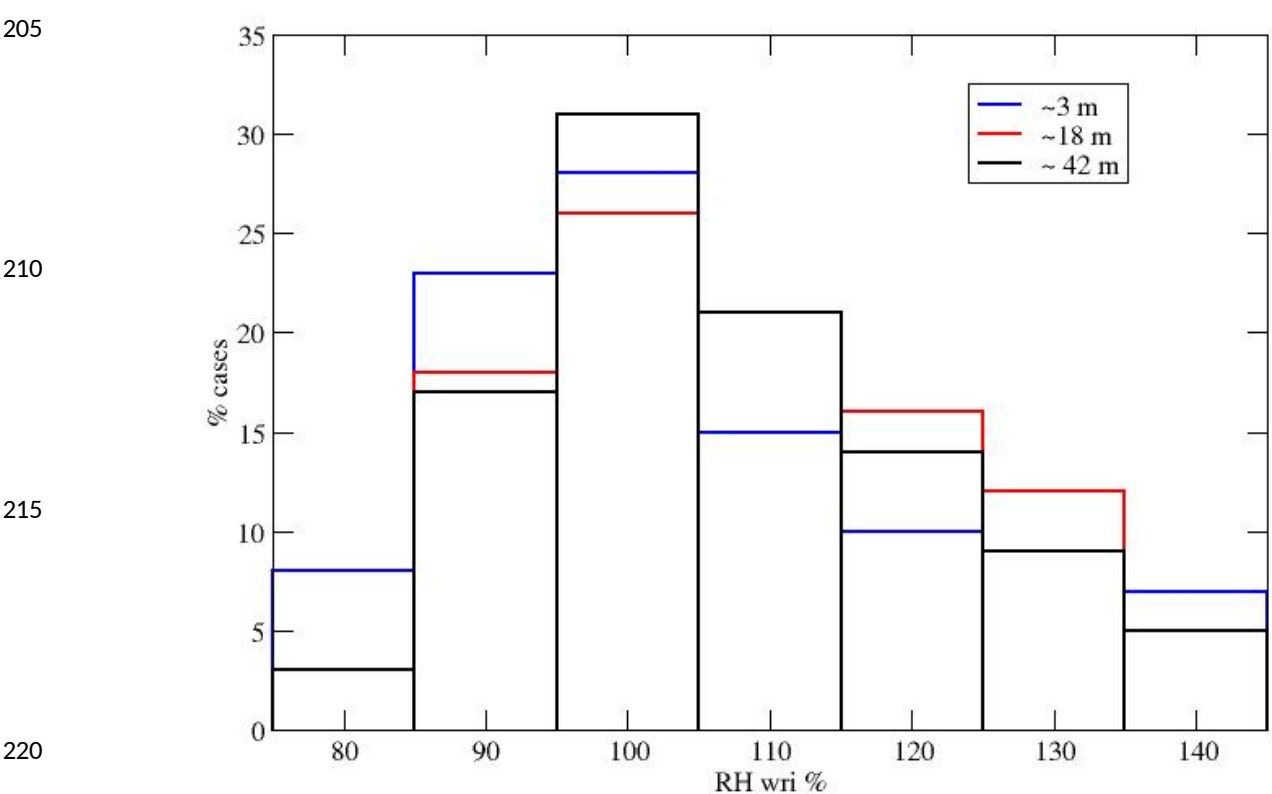

*Figure 5: Frequency distribution of  RHi observed at the 3 levels on the tower from 2018-2020.*



## 3.2 Seasonal and diurnal cycles and variability

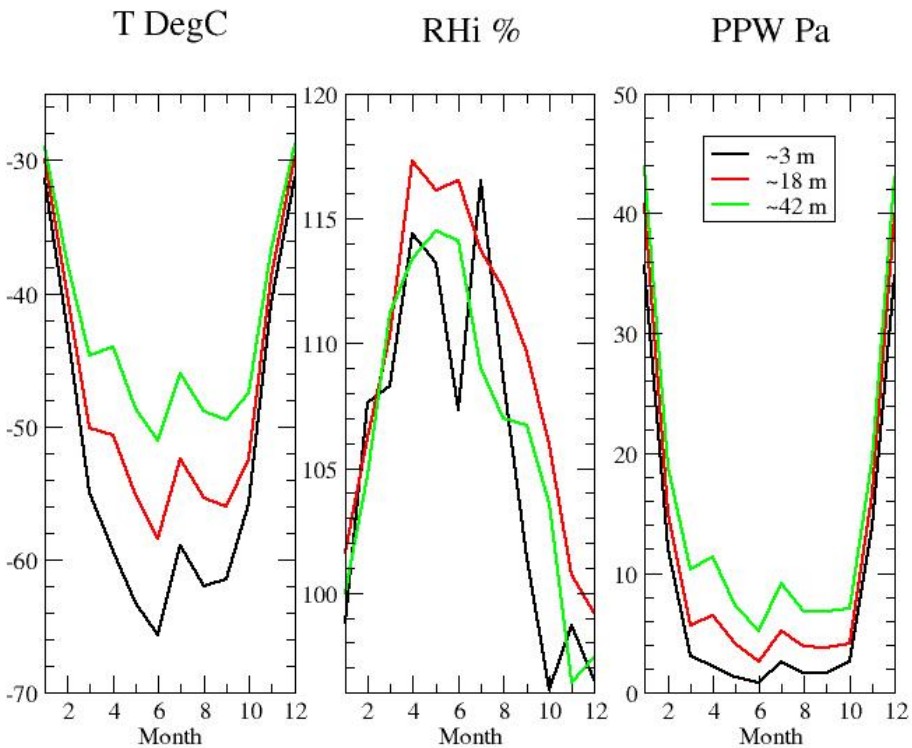

*Figure 6: Mean seasonal cycle of temperature, RHi and PPW at the 3 levels over the period 2018-2020.*

Beyond the fact that the atmosphere can be super or under saturated at Dome C, figure 3 indicates that temperature is a major controller of the water vapor content. As a consequence, one expects a strong diurnal and seasonal variability of PPW in response to the strong variations in temperature. The strong seasonal cycle is shown in figure 6, which also indicates that

RHi variation is opposite to and somewhat more confused than that for temperature and PPW. This is not unexpected as RHi is not a simple linear function of the two other variables. Both temperature and PPW show a relatively short maximum in summer and minimal values during much of the rest of the year. A local maximum of temperature and PPW is seen in July during the winter period, which might account for a sharp peak of RHi at the lowest level observed. However, for temperature, this does not appear when a 10-year average is used (not shown), and Figure 7 confirms that this local winter

maximum is within the natural interannual variability for this site.



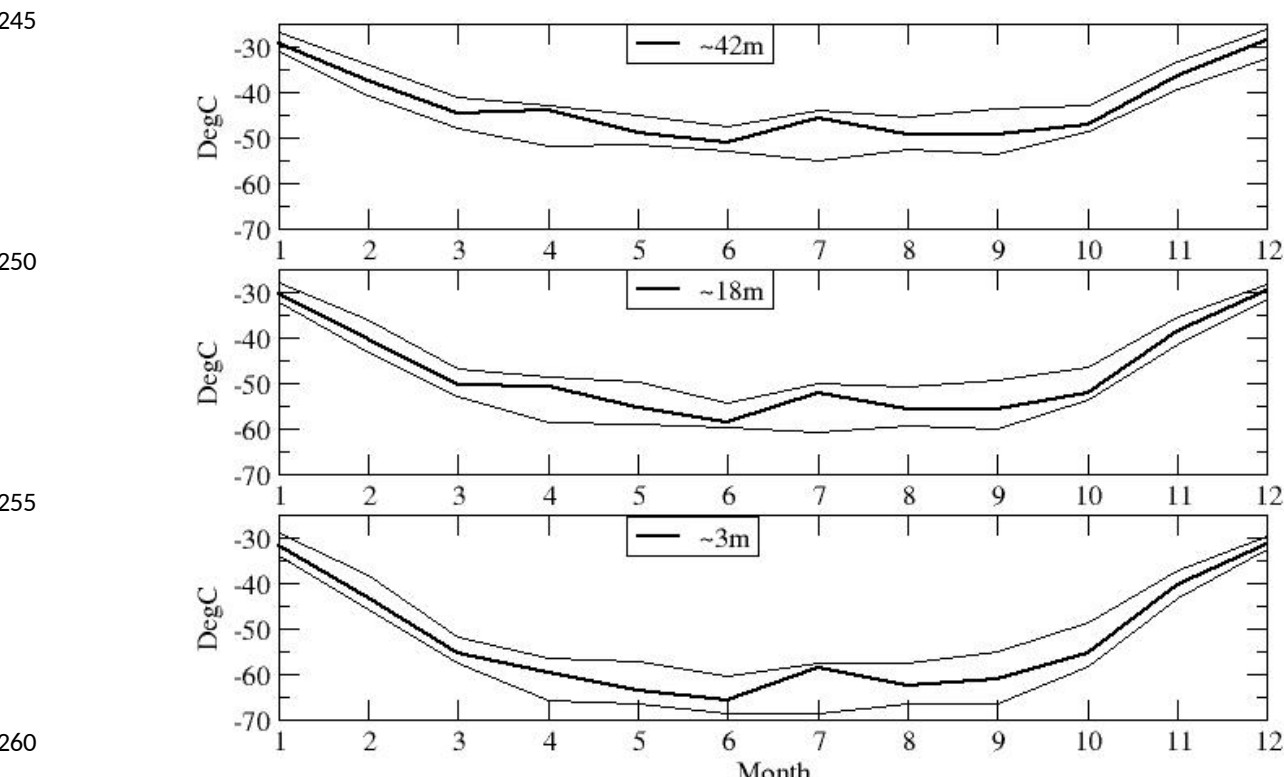

*Figure 7: Seasonal cycle of temperature averaged over 3 years (2018-2020) at the 3 levels with paired, modified thermohygrometers (bold line). The thin lines show the envelope of the ±2standard deviations of the interannual variability from the 10-year (2010-2019) seasonal cycle published in Genthon et al. (2021).*

Figure 8 shows the mean 24-hour cycle of temperature, RHi and PPW in December (full summer, left column)) and June (full winter, right column). Here, "24-hour cycle" does not necessarily refers to a diurnal solar cycle since there is no forcing of a diurnal cycle during the winter night. This is well illustrated by the plots in the right column at of figure 8, showing little variation over 24 hours. On the other hand, winter is when vertical gradients are largest. Then, while temperature increases by more than 15°C across 40 m height, PPW increases by a factor of 5. The increase with height is smooth for temperature and PPW, but not for RHi, for which much of the increase with height is between the 2 lower levels. Between the 2 upper levels RHi decreases in winter, while both mean temperature and mean PPW continue to increase. The combined vertical structure of temperature and moisture therefore deserves special attention.



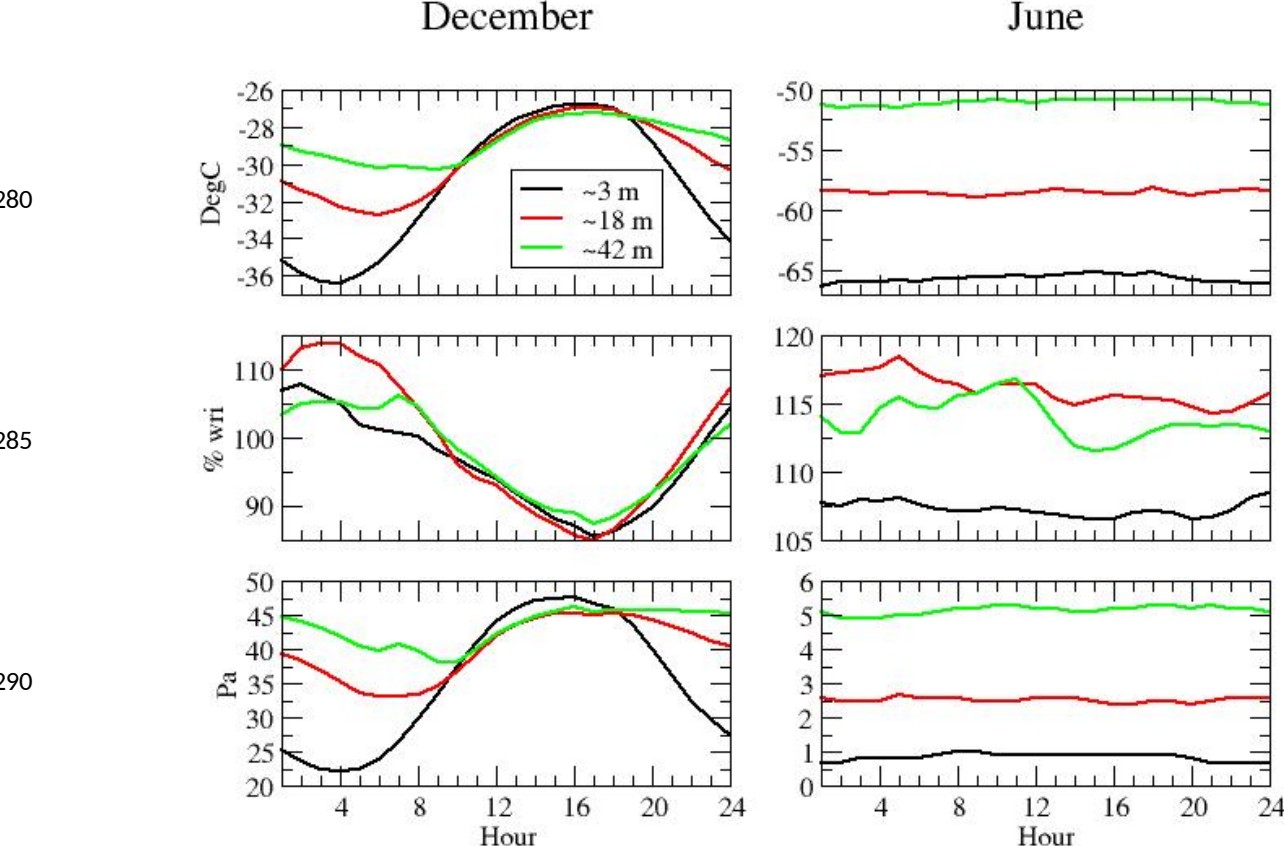





*Figure 8: Mean 24-hour cycle of temperature (upper plots), RHi (middle plots) and PPW (lower plots) in December (at left) and June (at right) averaged over 2018-2020.*

## 3.3 Vertical profiles and gradients


Figure 9 presents the annual average vertical profiles of annual temperature, RHi and PPW averaged for the 3-year data set. Unsurprisingly, lower PPW is associated with colder temperature near the surface. Both temperature and PPW show monotonically increasing values with height. This is not the case for RHi, which is consistently above 100% but for which

the maximum is at mid-level, with similarly lower values at top and bottom of the observed atmospheric layer. The non-linear dependence of relative humidity on temperature and atmospheric water content is responsible for this. Non-linearity also affects temporal averaging, and so to illustrate the vertical profile characteristics, a single case (26 July 2020, 05:00



local time) is presented in figure 10 showing the same feature. Humicap-based sensors such as the HMP155 report the relative humidity rather than atmospheric moisture content. Thus, at a site like Dome C, the source data may at first sight

appear peculiar because of the non-monotonic profile, but this feature is correct. The vertical profile of moisture content, here characterized by PPW which directly relates to concentration since the total pressure varies negligibly over 40 m, determines the turbulent fluxes. On average, the PPW gradient is positive throughout the sampled air layer, but it varies in time and can switch direction.


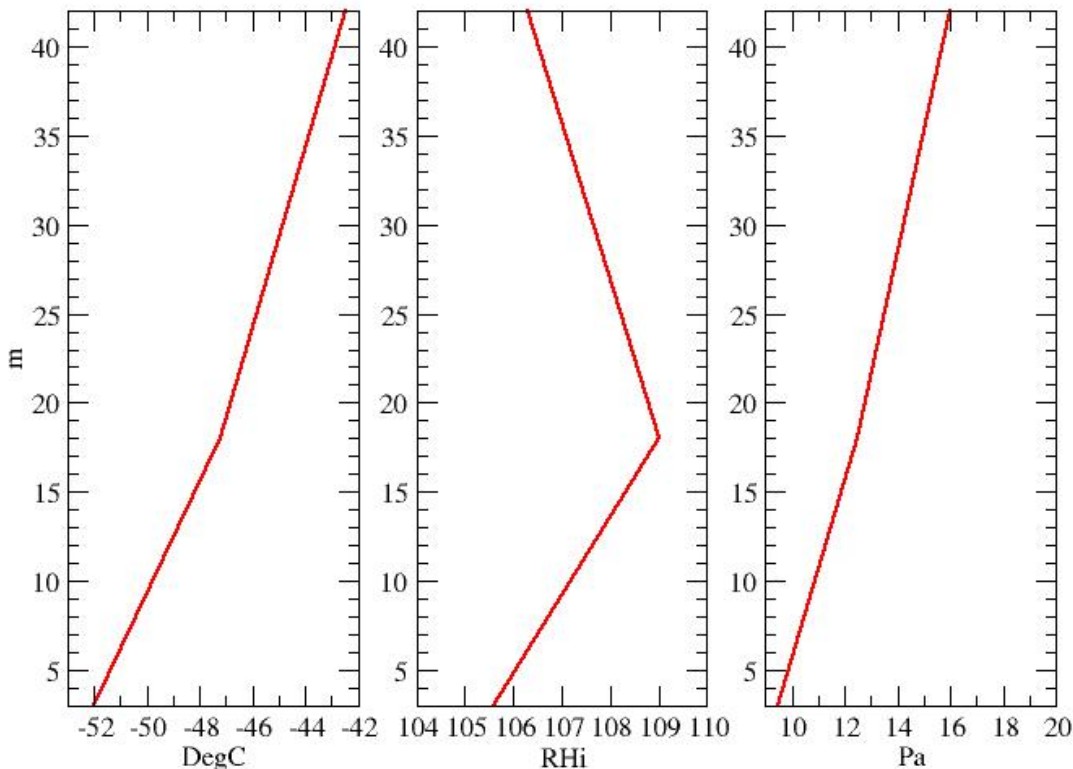

*Figure 9: Annual mean vertical profile of temperature (left), RHi (middle) and PPW (right) for observations made in 2018-2020.*




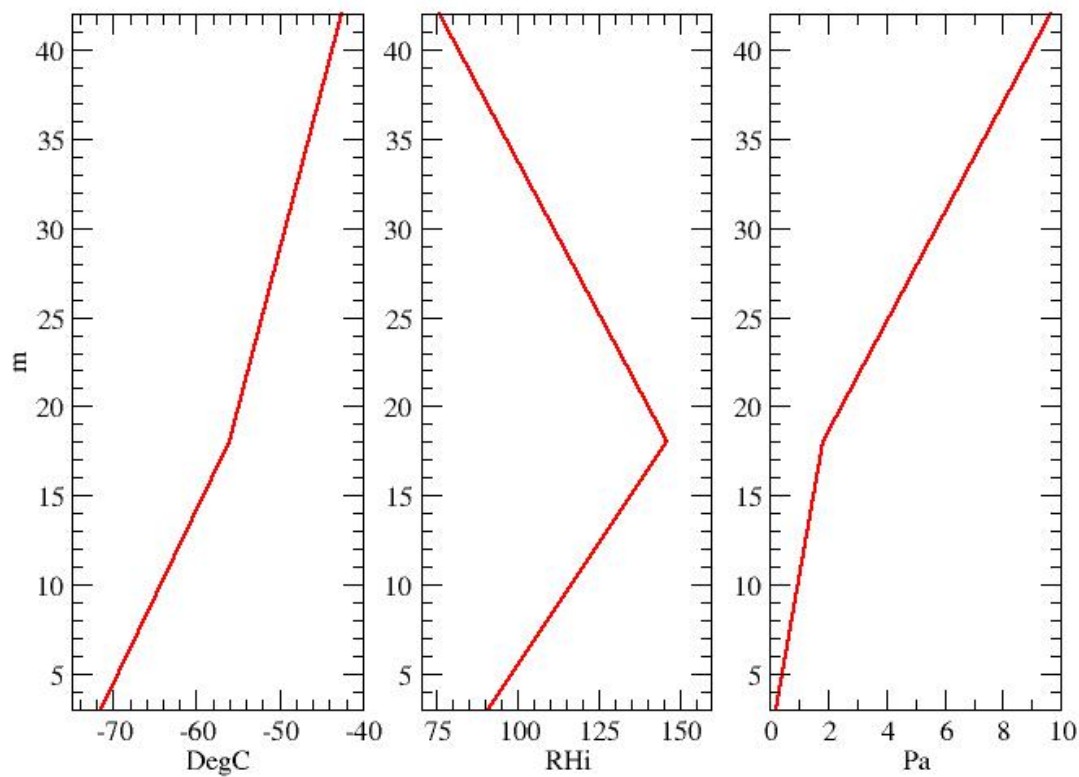

*Figure 10: Vertical profile of temperature (left), RHi (middle) and PPW (right) for 26 July 2020 05:00 local time.*

Correlations of high-frequency vertical wind speed and moisture fluctuations are necessary to calculate the amplitude of the
vertical moisture flux in the boundary layer but models parameterize the fluxes as a function of vertical gradients. Estimating
the fluxes themselves is beyond the scope of the present data paper. On the other hand, the gradients are further described
here because they provide a preliminary source for evaluating the ability of models to parameterize fluxes: if the gradients
are wrong in the models, then the fluxes are unlikely to be correct, and vice versa. In 89.6% of the time samples from the
observed period of 2018-2020, the gradient over the observed range of heights has the same sign as the mean gradient shown
in figure 9: positive upward, which results in a positive downward turbulent moisture flux ( i.e. the atmosphere "feeds" the
surface snow pack). Figure 11 shows that even though less frequent in summer than in winter, a positive gradient upward is
most frequent in all seasons, even in summer when warmer temperature may favor the sublimation of surface snow.
However, in December and January, the gradient is inverted more than 1/3 of the time. When this occurs, the water vapor
flux is downward, exporting surface sublimated moisture. Over the rest of the year, this occurs a small fraction of the time,

but because this is when PPW is largest (Figure 6) the accumulated net impact on the moisture budget in the surface atmosphere may be large.

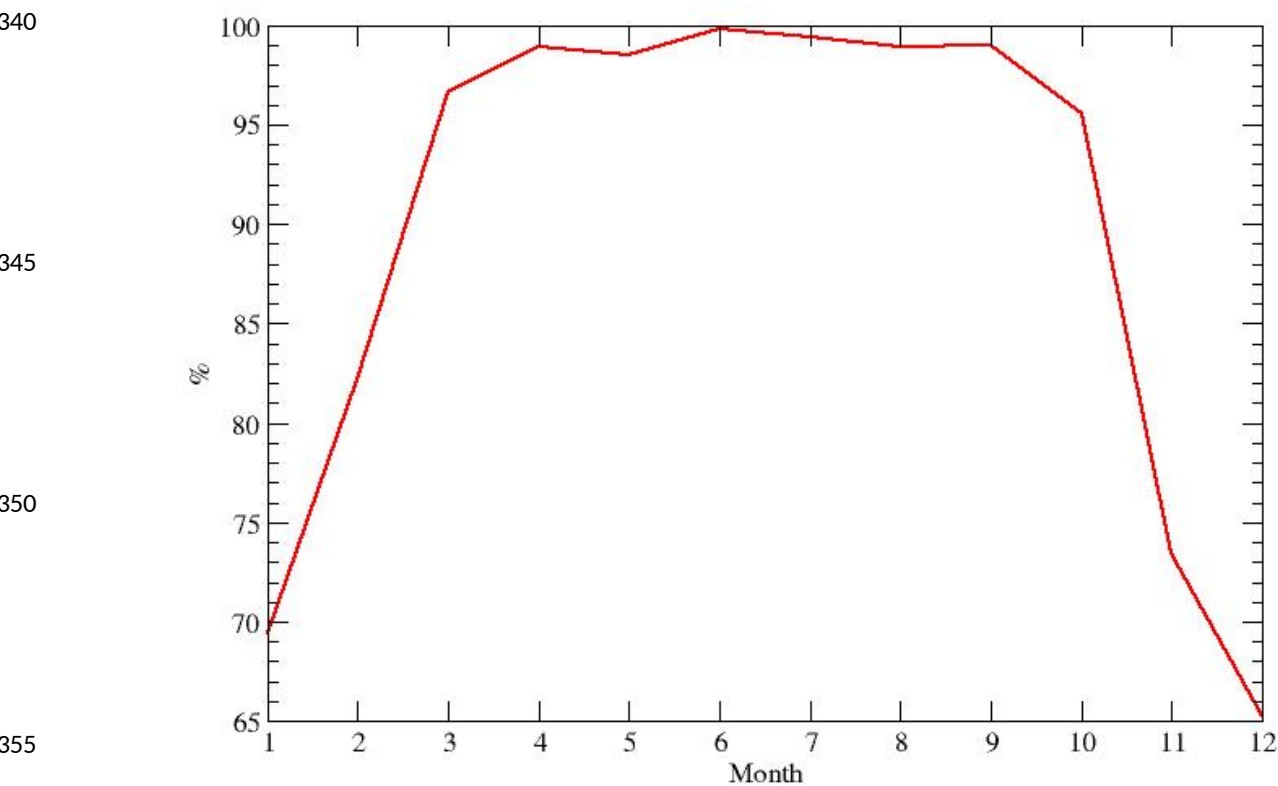

*Figure 11: Monthly frequency of cases of an upward vertical gradient of PPW, implying downward turbulent moisture flux.*

For the same reason, one expects that in summer, a strong diurnal cycle of temperature is associated with a strongly varying vertical gradient of atmospheric moisture (figure 8). On average in December and January, the gradient is positive upward more than 67% of the time. Figure 12 shows that in December and January this is very frequent during the coldest "night-time" hours (although there is no real solar night at this time of year). Cases of a downward positive gradient (upward turbulent flux) only occur significantly during the peak of the local day, yet even then the gradient is positive downward (upward moisture flux) more than 60% of the time. Again, this is when the atmospheric moisture content may be largest, fed



by surface evaporation and vertical mixing by thermal convection (Genthon et al. 2021). In fact, figure 8 shows that, in the

370 local afternoon in December, PPW differs marginally at the 3 levels: the vertical gradient is very small because mixing and thermal convection occur. Turbulent mixing and shallow convection are often treated separately in meteorological and climate models, with distinct parameterizations. However, the observed profiles of temperature can be used to characterize shallow convection (when temperature is vertically homogeneous) and test models but for thermal convection, observations of atmospheric moisture provide limited added value.

375

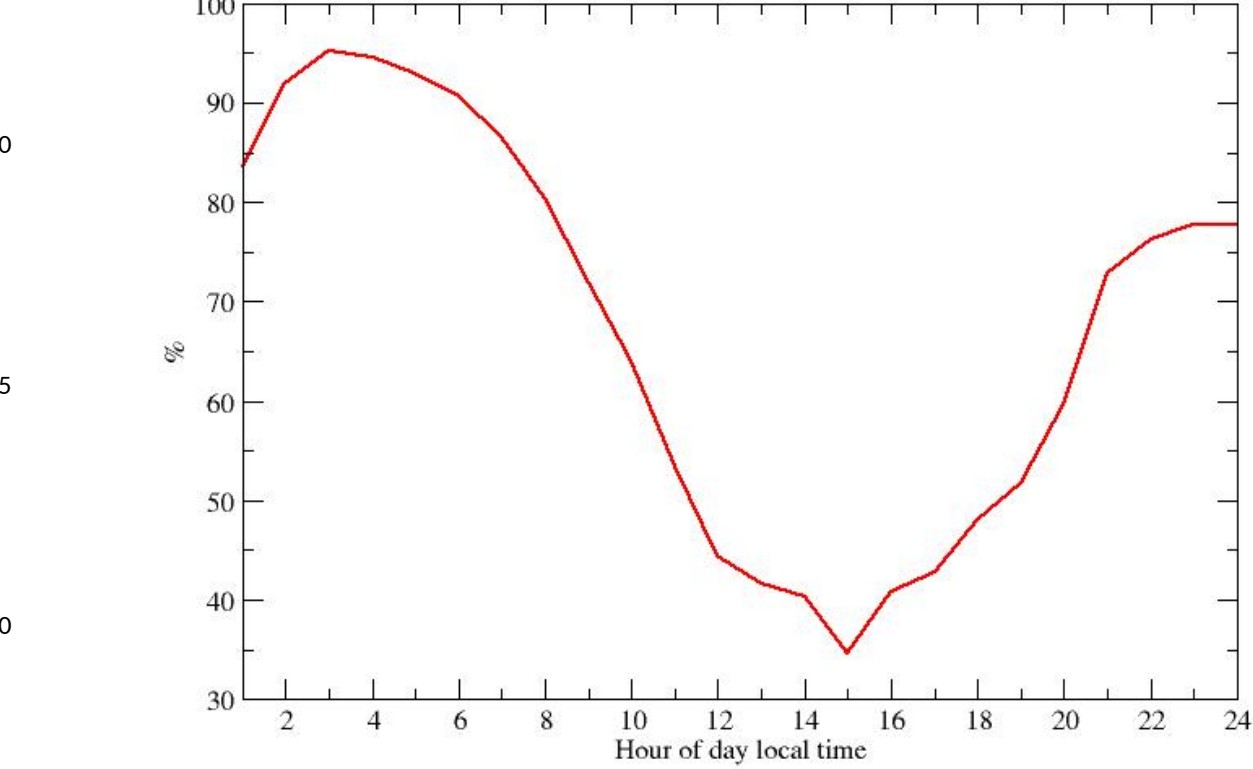

*Figure 12: Hourly frequency of cases with an upward vertical gradient of PPW, implying downward turbulent moisture flux in December and January (austral summer), for 2018-2020.*

**4. A look at relative humidity with respect to liquid water**



HMP155 sensors report relative humidity with respect to liquid water (Section 2) but this is subject to the limitations related to supersaturation with respect to ice as discussed above. In practice, unless heated, such a sensor cannot report relative humidity wrl above that corresponding to saturation with respect to ice. The heated HMP155 in the sensor pair shown in figure 1 is not affected by this, but is at a temperature higher than the environment. Just as for RHi, RHl at ambient
temperature can be calculated by combining the temperature and moisture by the heated HMP155 and temperature reported by the unheated HMP155 (Figure 1) using the GG formulae. Just like supersaturation wri is possible even if in principle thermodynamically unstable, supercooled water and liquid water clouds can exist at temperatures well below 0°C (Kenneth et al. (1985), Listowski et al. (2019), Ricaud et al. (2020a)). One may expect supercooled water to also occur in the surface atmosphere at Dome C and was actually reported at higher levels using remote sensing techniques (Ricaud et al. 2017). In
fact, the occurrence of supercooled water implies that the air is supersaturated with respect to ice.

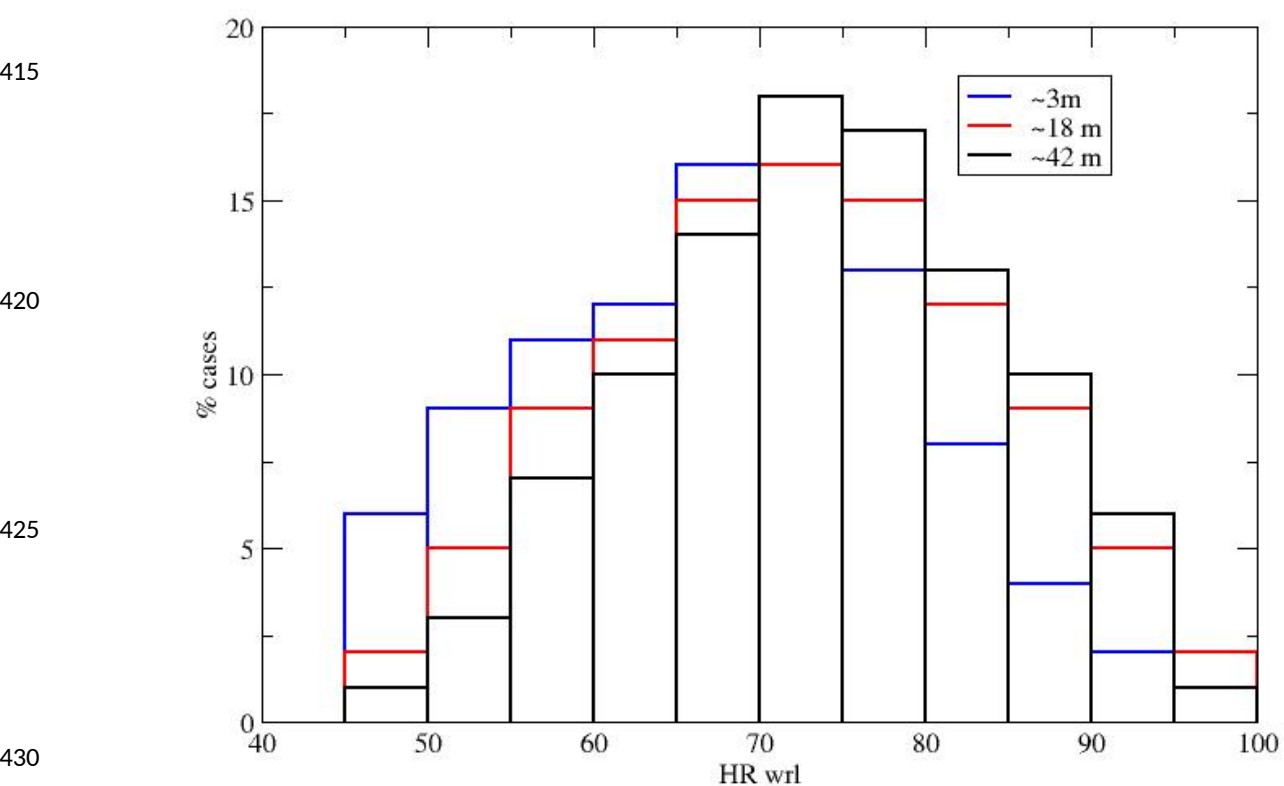

*Figure 13: Frequency distribution of occurrences of RHl at the 3 levels on the tower for 2018-2020.*





Figure 13 displays the frequency distribution of RHl as calculated using the observations reported here and GG relations. RHl hardly reaches or exceeds 100% (0.3% of the time), which is most likely due to the limited accuracy of the instruments and uncertainties on the saturation vapor pressure wrl at very cold temperatures (Murphy and Koop, 2005). The vertical gradient of RHl inverts as RHl increases, from downward to an upward gradient. The largest RHl occurs at mid-level, possibly due to the non-linearity of the relation between saturation RHl and air temperature and moisture content, just like

for RHi (Figure 10). Saturation or even slight supersaturation wrl can occur due to the surface tension of liquid droplets if water is present in the liquid phase. Figure 13 shows that this rarely occurs. Instead, one may expect that as RHl nears 100%, the water condenses and forms a cloud. While in the field deploying and attending to the instruments during the austral summer, the 1st and 2nd authors observed regular occurrences of surface haze occurring in the early morning then progressively vanishing after ~06:00 LT as the temperature progressively rose from the night minimum. This is quite

coherent with the few days in January 2020 shown in figure 14, during which RHl regularly increases in the evening as temperature cools and reaches a maximum in the following early morning which is slightly lower than 100%. That RHl does not quite reach 100% before condensation may be due to instrumental and conversion inaccuracies. It may also reflect that moisture is not homogeneously distributed and that it may locally reach 100% even if not at the site where the measurement is made, then triggering condensation at the larger scale.




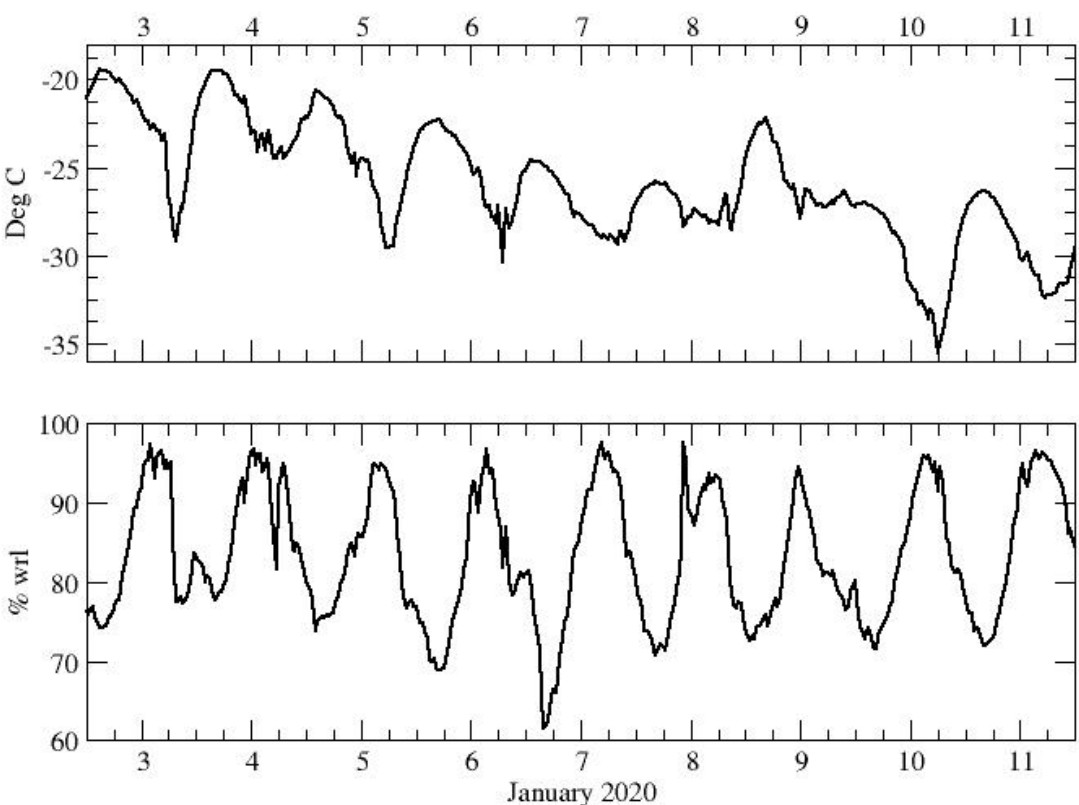

*Figure 14: A 10-day record of temperature and RHl at mid-level on the tower from January 2020.*

**5. Conclusions**

For the first time, a quasi-continuous multi-year record of atmospheric moisture in the surface atmosphere of the high
Antarctic Plateau has been obtained using instrumentation that can sample supersaturation in extreme cold conditions. This
dataset not only documents the temporal (seasonal, diurnal and synoptic) variability but also the vertical distribution of
moisture in the atmospheric layer from ~ 3 and ~42 m above the surface. One major signature of the high Antarctic Plateau





atmosphere is that the near-surface atmosphere, more often than not, is supersaturated with respect to ice. Most standard hygrometers cannot measure supersaturation; rather, the observations are capped at 100% when in reality RHi can reach well above. Therefore, it is likely that most observations of atmospheric moisture in the surface atmosphere of the high eastern Antarctic Plateau are biased dry. The vertical profiles of temperature and moisture partial pressure are generally monotonic,

albeit of variable sign. However, the vertical profile of RHi is generally non-monotonic and often has a maximum at mid-level, around 18-m in the observed period. This may be related to the non-linearity of the relationship between temperature and saturation humidity, and thus relative humidity. The vertical gradient of moisture is generally upward in winter (implying turbulent transport toward the surface, if any). The turbulent transport fluctuates with the diurnal cycle in summer, upward (implying downward turbulent flux) during the "night" (coldest part of the permanently sun lighted day) and weakly

downward during the "day".

Most modern meteorological and climate models now implement cold microphysics parametrizations that allow supersaturation as this is needed to correctly simulate high tropospheric clouds such as cirrus. In terms of cold temperature and high cleanliness (low levels of aerosols that can serve as CCN or IN), there are similarities between the high troposphere

and the elevated, near-surface atmosphere over the eastern Antarctic glacial plateau. Thus, it is no wonder that the parametrizations also produce supersaturation in the surface atmosphere at Dome C. Carrying out observation in this region, taking advantage of a permanently staffed Antarctic station, may not be completely straightforward but it is obviously much easier than making continuous moisture measurements in the high troposphere. The observational set up described in this paper, and the comparatively long time-series of atmospheric moisture including supersaturation may be used to evaluate and

improve parametrizations of cold microphysics.

In summer, visual observations and METARs (METeorological Aerodrome Report) indicate that haze/fog occasionally develops near the surface, then vanishes as temperature warms. A dedicated visibility sensor at mid-level on the tower (where RHi is on average largest, Figure 9) would be useful to study the processes of how condensation finally occurs in the

early morning after RHi has progressively increased well above 100% with night-time cooling (Figure 8). Beyond visibility, in situ observations of hydrometeor phase and size would add much value and should be considered, but are very challenging to acquire and difficult to ensure good quality in the extreme environment of the high Antarctic plateau.

Finally, variability and trends of temperature and atmospheric moisture have been recently reported at Dome C using

radiosondes, reanalyses and surface remote sensing instruments (Ricaud et al. 2020b). This is crucial information to better understand the links between Antarctica and the rest of the world in a changing climate. However, at Dome C, radiosondes are launched once a day, at 20:00 LT. In summer, this corresponds with the development of the nocturnal inversion (Genthon et al. 2021), thus a rapidly changing meteorological environment. In addition, while traveling across the first tens of meters above the surface, radiosondes may not be consistently well equilibrated with their atmospheric environment (Genthon et al.



2010) particularly with respect to moisture as the time response of the Humicap increases with colder temperature. Microwave remote sensing from the surface (Ricaud et al. 2010) is calibrated using the radiosondes, which is only for a small sample of the time and subject to possible radiosonde deficiencies as described above. Meteorological analyses and re-analyses are often questionable very near the surface because the observations available for assimilation are sparse and the parameterization of the very stable boundary layer in the models is often questionable (Bazile et al. 2014). In situ continuous

measurements would provide an important source of comparison for these less direct sources of information over a common time period. It is thus crucial to extend the time-series of continuous in situ meteorological observations as much as possible to best characterize common weather events, variability and trends over all time scales of interest and compare among data sources. It is hoped that the observation system described here will be supported to provide insight into the processes occurring in this extreme environment and provide a source of comparison for models and remotely sensed data, which

contributes significantly  to the science of meteorology, climate, climate variability and climate change in Antarctica.

**Data availability:**

The data presented here are made available on the PANGAEA open data repository at
https://doi.pangaea.de/10.1594/PANGAEA.939425 (Genthon et al., 2021).

**Acknowledgements:**

Logistical and some financial support for instruments and field work to deploy and service the observing system described here are provided by the French polar institute IPEV as part of project CALVA 1013. IPEV and the Italian Antarctic program PNRA jointly operate Concordia station at the Dome C. INSU (Institut National des Sciences de l'Univers) and OSUG (Observatoire des Sciences de l'Univers de Grenoble) also provide financial support to the CALVA project as a contribution to the GLACIOCLIM / CRYOBSCLIM observatory.


**Authors contribution**

CG leads the CALVA project, collected and processed the data used here, organized scientific discussions, wrote the bulk of the paper.

DV participated in several field campaigns, lead one, to obtain the data presented here, participated discussions to interpret

the data and to write the paper

EV and JBM participated field work, provided expertise in cold microphysics, participated discussions to interpret the data and to write the paper





LP designed and assembled the 3 improved systems (figure 1) that make possible to measure atmospheric moisture at Dome C including supersaturation and thus made possible the data set presented here.


**Competing interests:**

The authors declare no competing interests.

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
