# Peer review of "Water vapor in cold and clean atmosphere: a 3-year data set in the boundary layer of Dome C, East Antarctic Plateau"

_Earth System Science Data, 2021_

## Author Comment (AC1)

**Review of "Water vapor in cold and clean atmosphere: a 3-year data set in the boundary layer of Dome C, East Antarctic Plateau", by Genthon et al (ESSD-2021-414)**

(Our answers in red)

**General**

This paper documents an important new dataset of near-surface humidity profiles from Dome C on the East Antarctic Plateau, obtained using a novel hygrometer previously described by Genthon et al, 2017. Unlike many conventional hygrometers, the new instrument is capable of accurately measuring humidities which are supersaturated with respect to ice. Since ice supersaturation occurs frequently at Dome C this dataset provides for the first time an accurate climatological description of near-surface humidity profiles and their variability on daily to annual time scales at this station.

The paper provides a clear description of the measurements and the means by which they were obtained. Some basic climatological analysis is presented, which gives useful insight into the processes that control humidity at Dome C. I have a few comments on the paper (set out below) that require attention but these are all relatively minor and should all be easily addressed by the authors before final publication.

Thank you John King

**Specific comments**

1. Throughout the paper, the term "supersaturation" is used without qualification, and is generally used to imply supersaturation with respect to ice. There is thus some ambiguity in the use of this term and I would recommend that it should always be qualified by "wrt ice" or "wrt liquid water" as appropriate unless it is absolutely clear from the context which of these is being meant. Examples of places where clarification is definitely needed include the caption to fig.3, line 167 and line 473.

Agree. In several instances we use the term "supersaturation" to compactly express both senses, wrt ice and wrt water. In those cases we do not change the formulation. In other cases such as those specifically raised here, we reformulate to make it clear.

2. Lines 23-24: Make it clear that these temperatures are for Dome C, not averages for the whole plateau.

Done

3. Lines 78-79: Strictly speaking, the humidity gradient isn't the origin of the turbulence (which is generated by wind shear or convection). Maybe say "...because it enables the calculation of vertical moisture transport and exchange with the surface."?

Formulation is changed to "because gradient is the origin of the fact that turbulence can transport moisture vertically and exchange with surface"

4. Line 163: I'm not sure that this is the "traditional" view. Cloud physicists have known for a long time that ice supersaturation occurs in mixed-phase clouds - it is the basis of the Bergeron-Findeisen process, formulated in the 1930s. However, the occurrence of near-surface supersaturation wrt ice seems to have been largely overlooked until appropriate measurements (King and Anderson, 1999; Genthon et al, 2017) became available).

Right, here meaning, the traditional view in general, outside the specific cloud community which is "at the cutting edge" in the respect. Even in the climate modeling community, just 30 years ago, clouds were parameterized when water content reaches above 100%: there was no provision at all for supersaturation. This is still the case for some models including some participating in IPCC. But yes, this is clearly an outdated point of view. Here "traditional" replace by "old-time".

5. Figure 8: Is "Hour" local time? Give the difference to UTC.

Yes local time, which is UTC + 8, now reported in the legend.

6. Figure 9: There is very little information on this figure. You could make it more informative. Maybe show seasonal mean profiles?

We suspect that this comment is about figure 10, as an other reviewer made a similar comment on figure 10. The figure is now removed, the information being now conveyed in the text ("we looked at individual times to confirm it is not a result of this averaging")

7. Lines 326-327: King et al (2001, DOI: https://doi.org/10.3189/172756501781832548) present estimates of water vapour fluxes for Halley from humidity profiles.

OK but not sure what to do with this info. This sentence is to highlight what models do, not to discuss possible methods to calculate fluxes from gradients.

8. Lines 373-374: Not sure what you mean here - that water vapour profiles cannot be used to diagnose convection?

We mean that the information of water vapor profiles is no added value over the temperature gradient to evaluate thermal convection. It does contribute when phase change significantly affect the temperature profiles (in moist convection) but it is not the case here.

9. Lines 475-476: "Thus, it is no wonder…" Do you have a reference that shows that models with appropriate microphysical parametrisations do produce ice supersaturation near the surface at Dome C?

Yes, this is in Genthon et al. 2017, section 4.2 is dedicated to just this. Genthon et al. 2017 is cited.

10. Section 4: Also using a heated hygrometer, King and Anderson (1999) observed frequent ice supersaturation at Halley but no saturation wrt liquid water. They suggested that this indicated that, even in the clean polar air, cloud condensation nuclei, which will initiate droplet formation at very low supersaturations wrt liquid, are relatively abundant, while ice nucleating particles are rare.

OK this is now reported in the text

**Minor points and typographical corrections**

1. Line 76: "operational"

OK

2. Line 115: Capital "D" for Dome C

OK

3. Line 144: Delete space in "humidity"
OK

4. Line 183: "divergence" (not convergence)

It depends on point of view, here referring to convergence in the free atmosphere, admittedly associated with divergence at the surface

5. Lines 308-310: I don't understand this, please clarify.

This is slightly rewritten and hopefully now clarified. We mean that he native relative humidity data (such as distributed with the paper) show a profile similar to the middle plot of figure 9. It takes to to transform to partial partial to see that the mean profile is actually conventional

6. Line 335: "upward" (not downward)

OK

7. Lines 467-470: Insert "increasing" before "upward" and "downward"

We don't think that this is correct: the gradient (vector) is oriented upward or downward, not (necessarily) increasing in any direction.

---

## Author Comment (AC2)

**Review of "Water vapor in cold and clean atmosphere: a 3-year data set in the boundary layer of Dome C, East Antarctic Plateau" by Genthon et al.**

Our answers in red

This manuscript describes humidity data generated from a tower over Antarctica. This is an important data set in a unique environment. The authors do a good job of describing the relevance of the data. The manuscript is generally well written. It spends most of its time doing basic analysis of the data, with some interesting results. It is quite curious that there is no supersaturation with respect to liquid seen, even if the authors imply it from qualitative observations of possible droplets. I have a few major concerns, and minor comments.

Thanks

My major concerns are:

1. It seems odd to me that the manuscript pays more attention to analysis than to description of the sensors, calibration and errors. That seems appropriate for ESSD, while the scientific analysis perhaps belongs somewhere else. Section 2 should be expanded with more detail on the error characterization.

The description of sensors, calibrations and errors is admittedly limited because the sensors proper are commercial ones (HMP155). Thus basic information on calibrations and accuracies is provided by the manufacturer (Vaisala) and merely reproduced in the paper (section 2, lines 143-144). The dataset which is distributed on PANGAEA in association with the paper is the native data from HMP155, thus manufacturer information reported in section 2 directly applies and we feel there is no need to add much on this.

Section 2 also presents the methods to calculate elaborated data from the native HMP155 data. Obviously, this adds uncertainty on top of the native instrumental uncertainty. We suspect that this is the uncertainty the reviewer is interested in. This cannot be straightforwardly estimated as the added uncertainty results from the empirical formulations of the Clausius Clapeyron relations. There are several of them "on the market" and one way for an order of magnitude of the uncertainty would be to compare the results of different empirical formulations. However, Murphy and Koop report that results from those for the saturation vapor pressure over ice are all within 1% (except one which is not that used in the paper). This is now reported in the text. An additional intercomparison here would not add much.

In the next comment, the reviewer suggest that we use e. g. Murphy and Koop or show several different one, particularly for relative humidity over liquid. We now show and discuss relative humidity with respect over liquid using both GG and Murphy and Koop.

2. In particular, I think the paper should use the best available conversions for water vapor saturation pressure (e.g. Murphy and Koop 2005), or show several different ones. Particularly for Relative Humidity over Liquid.

Please see response to previous comment

3. In general the plot quality for line plots is not that great. As noted, some of the plots are redundant (Figure 7, Figure 10).

Yes, poor quality is due to converting native postscript images into jpg for inclusion in the Word document required to submit the manuscript. If / when the paper is accepted for publication, the optimal quality postscript images will be provided.

The july secondary maximum of temperature in the averaged seasonal cycle (figure 6) shows rather strongly and we think that figure 7 is needed to convince this is a result of interannnual variability in a relatively short record and has not climatological significance. Figure 7 has been reorganized and size reduced.

Figure 10 is removed

4. Some of the plots could be improved. The PDF historgrams at different heights (Figure 5, 13) would be easier to interpret with overlaid transparent filled colors for the bars.

5. There should be more discussion of possible errors in section 2 or in section 4. This seems strange given the discussion of potential supercooled liquid water, but no evidence of RH liquid > 100%.

Yes, there is room for possible errors and uncertainties in the discussions of data elaborated from the native instrumental reports, due to the elaboration process itself. We now somewhat discuss this issue in section 4 . However, a firm assessment of the errors is difficult here as, as Murphy and Koop themselves mention that formulations for vapor over solid are all within 1% and admit for the liquid the limit of any formulation due to "uncertainties on the saturation vapor pressure with respect to liquid at very cold temperatures", which are not quantified.

6. Links: not sure why the DOI refers to 3 more DOIs, one at each level. But I guess that is the Authors' choice.

The files were initially submitted as one 9-column file (plus time)  but PANGAEA suggested to make it 3 separate 3-column files (plus time), one for each level, . We abide with PANGAEA's suggestion, thus the 3 files.

Specific comments:

Page 5, L106: what is this reference to Genthon et al 2022: it's not in the references. Is it supposed to be the data itself?

Right, this should be 2021, now corrected

Page 6, L144: Does the Humicap calibration assume a saturation vapor pressure relationship? Seems like it must in the empirical calibration somewhere. Please elaborate and specify what is used or why it is not used.

Not sure what Humicap assumes but this is definitely an empirical relation, as already stated in section 5. Humicap implemented in HMP155 is used as "black box" here: it is up to Vaisala to devise calibration function, the end user merely measures a voltage which translates into relative humidity with respect to liquid even below 0°C. As a note, Humicaps are routinely used in similar black box mode in radiosondes worldwide.

Page 7, L163: Is the modified sensor in 3b the heated inlet on the right side of figure 1? Please clarify.

No, it is the full set up of figure 1. To correctly measure moisture, one needs both the unheated temperature and the heated temperature and moisture measurement. "instrument" replaced by "set of instruments"

Page 8, L167: does "adapted" mean "heated"?

No it means the full set, see above

Page 9, L181-8: Most of the discussion of mechanisms here is speculative. Do you have data that can bear on this? Figure 3b supports this figure, but not the mechanisms. Maybe show low winds with low temperatures? Or other measures of subsidence? Subsidence would lead to drying, but also warming, so pushing air out of those temperature ranges. Air aloft is going to have a higher potential temperature than surface Air, so without large radiative cooling it would be warmer.

Yes this is speculative. This is 1st of all a data paper. The discussion is meant to illustrate that the new data raise new issues; admittedly we do not solve all issues here. There is no direct measurement of subsidence as this is very slow: subsidence is mainly deduced from conservation issues (e.g. Vignon et al.). Subsidence may have a dominant impact on relative humidity only if lateral air and moisture advection are limited, that is, when the air is coldest: associating subsidence with warm(er than usual) air is thus not that straightforward.

Page 11, L240: Figure 7 doesn't really add new information. Couldn't you add standard deviations as colored shading to figure 6? Then you could do it for RHi and PPW too…

The standard deviation reported figure 7 is that over 10 years of temperature observation, to demonstrate that the July relative maximum has no climatological significance. Reporting on figure 6 could be misleading, as this is not the same time period, 3 years (fig 6) vs 10 years (fig 7). In fact, the reviewer suggests to show the same standard deviation for RHi and PPW as for temperature but this is not possible because we only have 3 years of data for the correct moisture variables. We thus feel it useful to separate what can be done over 10 years (temperature, figure 7) and over 3 years only (moisture), the 3 years of interest here (figure 6).

Page 15, L320: Figure 10. What does this figure show that is different from figure 9? I don't think this figure adds value to the paper and could be removed.

Following similar comment by reviewer 3, this figure has been removed

Page 18, L410: might want to note that since RhL < RHi then if RhL > 100% then RHi > 100%. That's the logic here right? If not, then how do you get supercooled liquid if RhL < 100%?

OK, we add that since RHl < Rhi, if RHl is close to 100% then RHi is necessary over 100%

Page 19, L445: Was the observed haze liquid or ice? That would seem to be an important distinction. Was there ever evidence of supercooled liquid at Dome C?

It is hard to assess whether near-surface haze is liquid or solid. We do not know of direct observation that can answer this question. On the other hand, at higher levels, there is definitely supercooled liquid water above Dome C as evidenced by polarized lidar observations ( Ricaud et al., 2020. Supercooled liquid water cloud observed, analysed, and modelled at the top of the planetary boundary layer above Dome C, Antarctica. *ACP* 20 (7), pp.4167-4191. ⟨10.5194/acp-20-4167-2020). Whith similar conditions within and above the boundary layer, one may assume that supercooled liquid haze is likely down to near the surface.

Page 19, L447: if you used a different saturation vapor pressure like Murphy and Koop would you get a different answer? That should reduce conversion inaccuracies.

We now use Murphy and Koop and we do get a similar answer as reported in the new text.

Page 19, L449: But earlier you argued that Dome C was pretty homogeneous? How does that match with the heterogeneity you imply here.

The site setting is very homogeneous on the large scale, the atmosphere is not necessarily on small scales, if only due to the sharp vertical gradients and turbulence from which any vertical mixing can induce local air inhomogeneity

Page 22, L508: This is the same repository and doi as Genthon et al 2021? Just checking that is supposed to be the case. What is the reference to Genthon et al 2022 noted above?

Yes, there was confusion here, corrected

---

## Author Comment (AC3)

**Review of "Water vapor in cold and clean atmosphere: a 3-year data set in the boundary layer of Dome C, East Antarctic Plateau" by Genthon et al. submitted to ESSD**

Our answers in red

The paper describes a 3 year timeseries of data from an Antarctic station tower with improved instrumentation to observe supersaturation with respect to ice. The discussion of the observations highlights the frequent occurrence of supersaturation with respect to ice at this location, occasionally up to water saturation. It is suggested this is a useful dataset for atmospheric model validation, with similar conditions to the upper troposphere.

This is an important and useful dataset for helping to quantify and understand supersaturated conditions and for evaluating models. The data is well described and the paper includes an informative analysis of the data.

Thank you

I have a number of comments where further clarity is required, particularly relating to gradients and turbulent fluxes, some of the Figures and use of supersaturation with respect to (wrt) ice or water.

**Comments**

In the "Short summary" for the paper, the last sentence is

"While supersaturation with respect to ice is frequent throughout the column, relative humidity with respect to (supercooled) liquid water reaches close to saturation."

As a one sentence summary of the results, I think this sentence, as it is, could be confusing.

Suggest something like:

"Supersaturation with respect to ice is frequently observed throughout the column, with relative occasionally humidities reaching water saturation."

Suppose this should be : "Supersaturation with respect to ice is frequently observed throughout the column, with relative humidities occasionally reaching water saturation."

OK taken, the short summary changed accordingly

Line 34

"and even more as freezing nuclei", but you mean even less? Suggest reword to make clearer.

OK, rewritten as: ."..and even fewer as freezing nuclei"

Line 41, Tomkins -> Tompkins

OK changed

Line 78

"the vertical humidity gradient … is the origin of turbulence". It's not the origin, do you mean that it is a result of the turbulence. Please reword this sentence to make it clearer.

OK, changed to "...is the origin of the fact that turbulence can transport…"

Line 93

Only use of "MO" is here and it is not defined. Suggest just state explicitly "Monin-Obukov"

Right, MO should have been mentioned line 82 as short for Monin Obukov. Now done.

Line 96-97

"the vertical moisture gradients used to represent the vertical distribution and mixing of moisture in the boundary layer". Again, a confusing statement. Please simplify/reword, e.g. "the vertical moisture gradient as a result of the mixing of moisture in the boundary layer".

This is not quite what we wish to express. This part is now rewritten as "the vertical moisture gradients which enter the parametrization of mixing of moisture in the boundary layer"

Line 115, dome C -> Dome C

Done

Line 117, according to Goff and Gratch

OK done

Lines 124-131.

There is a bit of duplication in this paragraph and it could be made simpler/shorter to improve clarity.

OK this is transitional text which is now shortened and wrapped up in just one sentence.

Line 129 -> saturation vapor pressure.

OK done

I don't think Figure 2 is necessary and would suggest removing it. The same curve (although on a linear scale) is in Figure 3.

We agree that the same the information on figure 2 shows also on figure 3 but not the same way and this is on purpose. We feel that the fact that expected values range over 3 orders of magnitude is best conveyed by figure 2 while figure 3 is best to show the actual data - the use of a modified y scale being explicitly mentioned in the legend of figure 3. We leave it to the editor to decide whether both figures are indeed useful or to remove figure 2, noting that none of the 2 other reviewers complain having the 2 figures.

Figure 3. For context for later discussions, you could also consider putting svp wrt water on the Figure, highlighting the upper bound of Rhi?

Thank you for this great suggestion. Taken, the new figure shows RH wrt liquid vapor which indeed nicely caps the data cloud.

Line 164, "by the CC curve"

Done

Line 160-180

Because of the previous discussion that the instrument reports vapor pressure with respect water, for clarity it is really important you are clear that Figure 3 and this paragraph discusses vapor pressure with respect to ice, e.g. on line 167, and "ice saturation" on line 178. Also mention this in the caption for Fig 3.

OK. Corrected for line 167, line 178 already explicitly mention "wri". Concerning fig 3, now explicited "Claussius – Calpeyron relations for relative humidity *wri*

Line 167, "frequently largely" -> "frequently"

We really mean that is is frequently **largely** above, not just frequently above.

Line 186, "adiabatic and radiative cooling combine"

Radiative cooling is more obvious, but could you say more about the conditions in this mid-temperature range that are leading to the adiabatic cooling (i.e. ascent) as this is perhaps less obvious over this region.

This is line 187. The observation site is at the summit of a dome which is locally very flat,. However, while the steepest surface elevation change (slope) and associated adiabatic cooling occurs within the 1$^{st}$ couple of hundreds of kms from the coast, the surface elevation still slowly rises further on the way to Dome C.

Line 197, the 18m level also does not show in the 105%-115% range. For improved clarity and simpler explanation I suggest showing the bars in a different way in Figure 5 so all can be seen - either offset or a set of three bars for each bin.

This is no longer an issue with the new figure of the results. However, to make it fully clear, different line types are now used in addition to different colors for the different levels.

Figure 7  - this is exactly the same data as used for the temperature data in Fig 6. Is it needed? Could the information be put on Fig 6 or just described in the text?

The same data are used for the average but more data 10 years are used to calculate the standard deviation. The standard deviation curves could have been implement on figure 6 but the results was somewhat messy and we elected rather to show std on a separate figure. It is expected that figure 7 will be reduced in size when a final version of the paper occurs.

Line 268, "mid" summer/winter rather than "full" summer/winter?

Yes, taken

Line 269, refers -> refer

Right

Line 270 , remove "at"

Right

Line 312, "vertical profile of moisture content determines the turbulent fluxes"

Again, surely the other way round, the vertical profile is determined by the turbulent fluxes. Or you mean it determines the sign of the transport due to the turbulent fluxes?

This is a chicken and egg issue. The gradient determines the flux, to the extent that whatever the quantity of turbulence if the gradient is zero the flux is zero, and with given turbulence the flux is proportional to the gradient. But then the flux in turn affects (reduce) the gradient so in some way the turbulence affects the gradient. We now write: with a given level of turbulence, the vertical profile of moisture determines the turbulent flux.

Does Figure 10 add much? You could just say you looked at individual times to confirm it is not a result of the averaging?

Following rev 2 and 3 comments, figure 10 is removed. We know simply mention that that we looked at individual times to confirm that this is not a result of averaging.

Lines 330, 332. Which gradient? Be clear you are talking about the PPW gradient in this case.

OK gradient "of PPW" added

Line 335, "water vapor flux is downward, exporting surface sublimated moisture"

Did you mean upward?

Yes, corrected

Line 372, for dry convection, surely potential temperature is the key quantity for instability, not temperature?

Yes but over such a shallow depth (~40 m) there is hardly a difference between gradients of temperature and of potential temperature. It is simpler to just mention temperature here.

Line 468,"diurnal cycle in the summer, upward.."

OK

Diurnal cycle of the moisture gradient…

Done

Figure 13, same comment as for Fig 5 for improved clarity.

Yes, again, different line types are now used in addition to different colors for the different levels